# REALTIME QUERY COMPLETION VIA DEEP LANGUAGE MODELS

## ABSTRACT

Search engine users nowadays heavily depend on query completion and correction to shape their queries. Typically, the completion is done by database lookup which does not understand the context and cannot generalize to prefixes not in the database. In the paper, we propose to use unsupervised deep language models to complete and correct the queries given an arbitrary prefix. We address two main challenges that renders this method practical for large-scale deployment: 1) we propose a method for integrating error correction into the language model completion via an edit-distance potential and a variant of beam search that can exploit such a potential function; and 2) we show how to efficiently perform CPU-based computation to complete the queries, with error correction, in real time (generating top 10 completions within 16 ms). Experiments show that the method substantially increases hit rate over standard approaches, and is capable of handling tail queries.

## 1 INTRODUCTION

Search completion is the problem of taking a prefix of a search query from a user and generating several candidate completions. This problem has enormous potential utility and monetary value to any search provider: the more accurately an engine can find the desired completions for a user (or indeed, potentially steer the user towards high-value completions), the more quickly it can lead the user to their desired goal.

This paper proposes a realtime search completion architecture based upon deep character-level language models. The basic idea is that instead of looking up possible completions from a generic database, we perform search under a deep-network-based language model to find the most likely completions of a user's current input. This allows us to integrate the power of deep language models, that has been shown to perform extremely well on complex language modeling and prediction tasks, with the desired goal of finding a good completion. Although this is a conceptually simple strategy (and one which has been considered before, as we highlight in the literature below), there are two key elements required to make this of practical use for a search engine provider, which together make up the primary technical contributions of the paper: 1) The completion must be *error correcting*, able to handle small errors in the user's initial input and provide completions for the most likely "correct" input. We propose such an approach that combines a character-level language model with an edit-distance-based potential function, combining the two using a tree-based beam search algorithm; 2) The completion must be *realtime*, able to produce high-quality potential completions in a time that is not even perceivable to the user. We achieve this by developing an efficient tree-based version of the error-correcting beam search, exploiting CPU-based computation for single queries (due to the high levels of branching in the beam search), and through numerous optimizations to the implementation that we discuss below.

We evaluate the method on the AOL search data set, a dataset consisting of over 36 million total search queries. In total, the method substantially outperforms highly optimized standard search completion algorithms in terms of its hit rate (the benefit of the deep language model and the error correction), while being fast enough to execute in real time for search engines. Experiments and code are all available online, and a real-time demo of the approach is available at `http://www.deepquerycompletion.com`.

## 1.1 Background on search completion

Here we review existing approaches to search query completion and error correction. Broadly speaking, two types of query completions are most relevant to our work, *database lookup* methods and *learning-based* approaches.

**Database Lookup** . One of the most intuitive ways to do query completion is to do a database lookup. That is, given a prefix, we can fetch all the known queries matching the prefix and return the most frequent candidates. This is called the "most popular completion" (MPC) (Bar-Yossef & Kraus, 2011), which corresponds to the maximum likelihood estimator for $P(\text{completion} \mid \text{prefix})$. The database lookup can be efficiently implemented by a trie (Hsu & Ottaviano, 2013). For instance, it takes only $15\mu s$ to give 16 suggestions for a query in our own trie-based implementation. However, due to the long tail nature (Szpektor et al., 2011) of the search queries, many prefixes might not exist in the database; for example, in the AOL search data, 28% of the queries are unique. An excellent survey of these current "classical" approaches is given in (Cai et al., 2016).

**Learning-based** . In addition to database lookup approaches, in recent years there have been a number of approaches that use learning-based methods for query completion. Sordoni et al. (2015) uses a translation model at the word level to output single-word search query suggestions, and also model consecutive sessions of the same user. Liu et al. (2016) proposed a word-based method for code completion, but focused solely on greedy stochastic sampling for the prediction. Mitra & Craswell (2015) also used neural networks combined with a database-based model to handle tail queries, but focused on CNN approaches that just output the single most likely word-level completion. Shokouhi (2013) used logistic regression to learn a personalized query ranking model, specific to individual users. All these approaches are relevant but fairly orthogonal to our own, as we focus here on character-level modeling, beam search, and realtime completion. Finally, Park & Chiba (2017) very recently published an approach similar to ours, which uses a character-level language model for completion. But their approach focuses on the use of embeddings (such as word2vec) to produce "intelligent" completions that make use of additional context, and the approach does not handle error correction; they also do not report the prediction time of their completions, which is a key driver for our work.

## 1.2 Error correction for queries

Our work also relates to methods on error and spelling correction approaches, which again are roughly divided into heuristic models and learning-based approaches.

**Heuristic models** Whitelaw et al. (2009) proposed generating candidate sets that contain common errors for given prefixes, then searching these based upon the current query. Similarly, Martins & Silva (2004) use a ternary search tree to accelerate the search within candidate sets for spelling correction in general. The approaches are nice that they are easily parallelizable at runtime, but are relatively "brute force", and cannot handle previously unseen permutations.

**Learning-based Model** On the learning side, Duan & Hsu (2011) train an $n$-gram Markov model combined with A* search to determine candidate misspelling; this is similar to our approach except with a much richer language model replacing the simple $n$-gram model, which creates several challenges in the search procedure itself. Likewise, (Xie et al., 2016) use a similar character-level model with attention, but do so in the context of error correcting an entire paragraph of text, and don't focus on the same realtime aspects that we do.

## 2 Background on Query Completion

When a user types any prefix string $s$ in the search engine, the query completion function will start to recommend the best $r$ completions, each denoted $\hat{s}$, according to certain metrics. For example, one might want to maximize the probability that a recommendation is clicked. The conditional probability can be formulated as

$$P(\hat{s} \mid s) := P(\text{completion} \mid \text{prefix}), \qquad (1)$$

and the goal of query completion in the setting is to find the top $r$ most probable strings $\hat{s}$ which potentially also maximize some additional metric, such as the click-through rate.

Denote $s_{0:m}$ as the first $m$ characters in string $s$ and $s_0$ as the empty set. We first discuss the query completion in a simplified setting, in which all completions must contain the prefix exactly, that is: $\hat{s}_{0:m} = s_{0:m}$, and

$$P\left(\hat{s}_{0:n} \mid s_{0:m}\right) = P\left(\hat{s}_{m+1:n} \mid s_{0:m}\right) = P\left(\hat{s}_{m+1:n} \mid \hat{s}_{0:m}\right) \tag{2}$$

where $n$ is the total length of a completion. Note that the probability is defined in the sequence domain, which contains exponentially many candidate strings. To simplify the model we can apply the conditional probability formula recursively and we have

$$P\left(\hat{s}_{m+1:n} \mid \hat{s}_{0:m}\right) = \prod_{t=m}^{n-1} P\left(\hat{s}_{t+1} \mid \hat{s}_{0:t}\right). \tag{3}$$

This way, we only need to model $P\left(\hat{s}_{t+1} \mid \hat{s}_{0:t}\right)$, that is, the probability of the next character under the current prefix. This is precisely a character-level language model, and we can learn it in an unsupervised manner using a variety of methods, though here we focus on the extremely popular approaches of using recurrent neural networks (RNNs) for this character-level language model. Character-level models are the right fidelity for search completion, because word-level models or sequence-to-sequence probabilities would not be able to model probabilities under all partial strings.

## 2.1 THE UNSUPERVISED LANGUAGE MODEL

Let's focus on the language model term $P\left(\hat{s}_{t+1} \mid \hat{s}_{0:t}\right)$, the probability of next character under the current prefix. RNNs in general, and variants like long short term memory networks (LSTMs) (Hochreiter & Schmidhuber, 1997), are extremely popular for high-fidelity character level modeling, and achieve state of the art performance for a number of datasets (Chung et al., 2016). Since they can be trained from unsupervised data (e.g., just datasets of many unannotated search queries), we can easily adapt the model to whatever terms users are *actually* searching for in the data set, with the potential to adapt to new searches, products, etc, simply by occasionally retraining the model on all data collected up to the current point.

Although character-level language modeling is a fairly standard approach, we briefly highlight the model we use for completeness. Consider a recurrent neural network with hidden state $h_t$ at time $t$. We want to encode the prefix $\hat{s}_{0:t}$ and predict the next character using $h_t$. We follow fairly standard approaches here and use an LSTM model, in particular the specific implementation from the Keras library (Chollet et al., 2015)[1], which is defined by the recurrences

$$i_t = \sigma\left(W_{xi}x_t + W_{hi}h_{t-1} + b_i\right), \tag{4}$$
$$f_t = \sigma\left(W_{xf}x_t + W_{hf}h_{t-1} + b_f\right), \tag{5}$$
$$o_t = \sigma\left(W_{xo}x_t + W_{ho}h_{t-1} + b_o\right), \tag{6}$$
$$c_t = i_t \odot \tanh\left(W_{xc}x_t + W_{hc}h_{t-1} + b_c\right) + f_t \odot c_{t-1}, \tag{7}$$
$$h_t = o_t \odot \tanh\left(c_t\right), \tag{8}$$

in which $h_t, b \in \mathcal{R}^d$, $x_t \in \mathcal{R}^{|C|}$, $\forall t$, $W_{xi}, W_{xf}, W_{xo}, W_{xc}$ and $W_{hi}, W_{hf}, W_{ho}, W_{hc}$ are the forward kernel and recurrent kernel with corresponding dimensions, and $\odot$ is the element-wise dot. We use a one-hot encoding of characters as input, a two-layer LSTM with 256 hidden units (more discussion on these choices below), and for prediction of character $\hat{s}_{t+1}$, we feed the hidden layer $h_t$ to a softmax function

$$P\left(\hat{s}_{t+1} = i \mid \hat{s}_{0:t}\right) = \text{softmax}\left(i; W_{\text{softmax}}h_t\right) = \frac{\exp(w_i^T h_t)}{\sum_{j=1}^{|C|} \exp(w_j^T h_t)}, \quad \forall i \in \text{character set } C \tag{9}$$

and train the language model to maximize the log likelihood (minimize the categorical cross-entropy loss),

$$\underset{W}{\text{minimize}} \ -\sum_{s \in S} \sum_{t=1}^{|s|} \log P(s_{t+1} \mid s_{0:t}), \tag{10}$$

---

[1]Note that, as we describe below, we won't actually use the Keras library at prediction time, but we do use it for training

where $S$ denotes the set of queries. Further, we pad all queries with an end-of-sequence symbol to predict whether the query is complete.

## 2.2 STOCHASTIC SEARCH AND BEAM SEARCH

Once we have the language model, we can evaluate the probability $P\left(\hat{s}_{m+1:n} \mid \hat{s}_{0:m}\right)$ for any completion $\hat{s}_{0:m}$, but would ideally like to find the completion with the highest probability. Enumerating all the possible strings is not an option because we have exponentially many candidates. Indeed, finding the best sequence probability, which is called the "decoding problem", is NP-hard(Forney, 1973), so we have to rely on approximation.

The most naive way to do so is simply via sampling: we sample the next character (according to its probability of occurrence) given the current prefix, until we hit an end-of-sequence symbol:

$$\textbf{For } t = m; \quad ; t\text{++} :$$
$$\hat{s}_{t+1} \sim P(\hat{s}_{t+1} \mid \hat{s}_{0:t});$$
$$\textbf{If } \hat{s}_{t+1}\text{== End-of-Seq : \textbf{break}};$$

This method produces output that looks intuitively reasonable. However, it is biased toward longer sequences with short-term dependencies and clearly does not generate the most probable sequences, because sampling in a greedy fashion clearly is not the same as sampling from the sequence space.

That is, we really need to do a better approximate search to get better results. One classic way to do this is to perform beam search, that is, perform breadth-first search while keeping the top-$r$ candidates. We illustrate the algorithm as follows:

$\text{cand} := \{s_{0:m} : 0\}, \quad \text{result} := \{\}$
$\textbf{For } t = m; \text{ cand is not empty}; t\text{++}:$
    $\text{cand}_{\text{new}} := \{ s_{0:t+1} : \log P\left(s_{0:t+1} \mid s_{0:m}\right) \text{ for every } s_{t+1}, \text{ for every } s_{0:t} \in \text{cand}\};$
    $\text{cand} := \text{the most probable } (r - |\text{result}|) \text{ candidates in cand}_{\text{new}};$
    Move $s_{0:t+1}$ from cand to result if $s_{t+1}$ is end-of-sequence symbol;

By performing beam search we can consistently obtain a more probable set of completions compared to stochastic search.

However, there are two issues with the above method. First, it does not handle error correction (which is necessary for any practical type of completion) since the completion always attempts to find sequences that fit the current prefix exactly. Second, as we show below, a naive implementation of this model is extremely slow, often taking on the order of one second to produce 16 completions for a given prefix. Thus, in the next two sections, we present our primary technical contributions, which address both these issues.

## 3 COMPLETION WITH ERROR CORRECTION

Most of the time query completion is more than completing over a fixed prefix. The input prefix might contain mistakes and sometimes we would also like to insert keywords in the prefix. Traditionally, the database community handles the two features by first doing a pass of error correction by matching the input to a typo database generated by permuting characters, then match the database again on the permuted terms for insertion completion. Our observation is that with a language-model-based approach, we can handle the spelling correction and insertion completion all in one model.

### 3.1 A POTENTIAL FUNCTION AS PROBABILITY CORRECTION

Remember that the original problem of estimating the probability of a query completion can be written as

$$P\left(\hat{s}_{0:n} \mid s_{0:m}\right). \tag{11}$$

Now, suppose that we no longer constrain the prefix to be exactly $\hat{s}_{0:m} = s_{0:m}$. To utilize the language model, we need to augment the conditional distribution by adding an additional probability

term $P(\hat{s}_{0:m'} \mid s_{0:m})$, the probability of a prefix in the completion given an observed prefix; note that these need not be the same length, as we may want to insert or delete characters from the prefix. Our completion probability now becomes

$$P\left(\hat{s}_{0:n} \mid s_{0:m}\right) = P\left(\hat{s}_{m'+0:n} \mid \hat{s}_{0:m'}, \; s_{0:m}\right) P\left(\hat{s}_{0:m'} \mid s_{0:m}\right) \tag{12}$$

$$= P\left(\hat{s}_{m'+0:n} \mid \hat{s}_{0:m'}\right) P\left(\hat{s}_{0:m'} \mid s_{0:m}\right), \tag{13}$$

where the last equality comes because we assume that the completion only depends on its prefix. The probability $P\left(\hat{s}_{0:m'} \mid s_{0:m}\right)$ models the error rate between the old and new prefix, that is, the probability of such modification/error of the old prefix. However, the question remains as to how to best represent this probability.

## 3.2 Amortized Dynamic Programming On the Search Tree

A natural candidate for measuring the distance between two strings is the edit distance function. Remember that the edit distance measures the minimum changes (add/remove/replace) to transform one string into another. If $\hat{s}_{0:m'}$ is the correct prefix, the edit distance between $\hat{s}_{0:m'}$ and $s_{0:m}$ can be interpreted as the number of errors in the original prefix. Assuming that the probability by which users make an error is constant, we can model the probability of spelling error as

$$\log P\left(\hat{s}_{0:m'} \mid s_{0:m}\right) = -\alpha \cdot \text{edit distance}(\hat{s}_{0:m'}, \; s_{0:m}). \tag{14}$$

Taking a 2% error rate gives $\alpha = -\log \frac{1}{50} \approx 4$. The edit distance can be calculated using the following dynamic programming (Wagner & Fischer, 1974), which we include here as we will shortly propose a modification that works better in the search completion setting:

$\text{dist}_{\text{edit}} := [0,1,\ldots,m];$

**For** $i = 0; i \le m'; i{+}{+}$**:**

    **For** $j = 0; j \le m; j{+}{+}$**:**

        **If** $\hat{s}_i{=}{=}s_j$**:**

            $\text{dist}_{\text{new}}(j) := \text{dist}_{\text{edit}}(j\text{-}1);$

        **Else :**

$$\text{dist}_{\text{new}}(j) = \min \begin{cases} \text{dist}_{\text{new}}(j\text{-}1) + 1, & \text{add;} \\ \text{dist}_{\text{edit}}(j\text{-}1) + 1, & \text{substitute;} \\ \text{dist}_{\text{edit}}(j) + 1, & \text{delete;} \end{cases}$$

    $\text{dist}_{\text{edit}} := \text{dist}_{\text{new}};$

  $\text{Output } \text{dist}_{\text{edit}}(m);$

The above algorithm takes $O(m \cdot m')$ time to run, and we need to evaluate the distance for every new candidate in the beam search. Thus, if we run it naively, it results in an additional $O(|C|rm \cdot m')$ overhead to the beam search procedure, where $C$ denotes the size of possible character set and $r$ is the number of completions. However, observe that every new candidate is extended from old candidates. That is, only one character is changed in the outer-loop of the edit distance algorithm if we can maintain $\text{dist}_{\text{edit}}$ for every candidates. By such the bookkeeping, we are able to amortize the edit distance algorithm over the search, resulting in a much lower $O(|C|rm)$ complexity.

## 3.3 Edit Distance v.s. Completion Distance

Note that to handle insertion completion, we should not incur penalty for adding words after the last character of any term. To accomplish this, we designed a new distance function called the "completion distance", which changes the update rule (15) to be

$$\text{dist}_{\text{new}}(j) = \min \begin{cases} \text{dist}_{\text{new}}(j\text{-}1) + \mathcal{I}(s_{j-1} \ne \text{last char}) & \text{add;} \\ \text{dist}_{\text{compl}}(j\text{-}1) + 1 & \text{substitute;} \\ \text{dist}_{\text{compl}}(j) + 1 & \text{delete;} \end{cases}$$

$$\tag{15}$$

By doing so, completion like "poke go" to "pokemon go" would not incur unnecessary penalties.

### 3.4 EXTENSIONS

Finally, we note that this idea of inserting a potential function between different prefixes naturally generalizes to contexts other than edit distance. For example, many product search engines wish to drive the user not simply to a high-probability completion, but to a completion that is likely to lead to an actual sale. By modifying the prefix probability to more heavily weight high-value completions, we can effectively optimize metrics other than simple completion probability using this approach.

## 4 REALTIME COMPLETION

Starting with the system as proposed previously, the key challenge that remains now is to perform such completions in real time. Response time is crucial for query completion because unless the user can see completions as they type the query, the results will likely have very little value. The bar we set for ourselves in this work is to provide 16 completions in 20 milliseconds on current hardware. Unfortunately, a naive implementation of beam search with the model trained above (using off-the-shelf implementations), requires more than one second to complete forward propagation through the network and beam search.

Thus, in this section we provide a detailed breakdown of how we have empirically improved this performance by a factor of over 50x, to achieve sub-20-ms completion times.

### 4.1 LSTM OVER A TREE

First, we observe that all new candidates in the beam search process are extensions from the old candidates because of the BFS property. In this case, the forward propagations would greatly overlap. If we can maintain $\boldsymbol{h}_t$ for every old candidate, extending one character for new candidates would require only one forward propagation step. That is, we amortize the LSTM forward propagation over the search tree. The algorithm is illustrated below.

$$\text{cand} := \{s_{0:m} : (\boldsymbol{h}_m,\ 0)\}, \quad \text{result} := \{\}; \qquad\qquad\qquad O(md^2)$$

**For** $t = m$**; cand is not empty; $t$++:**

$$\text{cand}_{\text{new}} := \{\ s_{0:t+1} : (\boldsymbol{h}_t,\ \log P\left(s_{0:t}\mid s_{0:m}\right) + \log P\left(s_{t+1}\mid s_{0:t}\right))$$

$$\qquad\qquad \text{for every } s_{t+1} \in C, \text{ for every } s_{0:t} \in \text{cand} \}\ ; \qquad O(r|C|d)$$

$$\text{cand} := \text{the most probable } r - |\text{result}| \text{ candidates in cand}_{\text{new}}; \qquad O(r|C|)$$

$$\text{Move } s_{0:t+1} \text{ from cand to result if } s_{t+1} \text{ is end-of-sequence symbol}; \qquad O(r)$$

$$\text{Bump } \boldsymbol{h}_t \text{ to } \boldsymbol{h}_{t+1} \text{ by one step of LSTM on } s_{t+1}, \forall s_{0:t+1} \in \text{cand}\ ; \qquad O(rd^2)$$

Note again that $m$ is the length of $s_{0:m}$, $d$ is the hidden dimension of LSTM, $|C|$ is the length of character set $C$, and $r$ is the number of completions required. Using this approach, the complexity for computing $r$ completions for $d$-dimensional LSTM reduces from $O(n^2 rd(d + |C|))$ to $O(nrd(d + |C|))$ for sequence with maximum length $n$. A naive C implementation shows that the running time for such search drops to 250 ms from over 1 sec.

### 4.2 CPU IMPLEMENTATION AND LSTM TWEAKS

Although GPUs appear to be most suitable for computation in deep learning, for this particular application we found that the CPU is actually better suited to the task. This is due to the need for branching and maintaining relatively complex data structures in the beam search process, along with the integration of the edit distance computation. Thus, implementation on a GPU requires a process that frequently shuffles very small amounts of data (each new character), between the CPU and GPU. We thus implemented the entire beam search and forward inference in C on the CPU.

However, after moving to a pure CPU implementation, it is the case that initially about 90% of the time is spent on computing the matrix-vector product in the LSTM. By properly moving to batch matrix-matrix operations with a minibatch that contains all $r$ candidates maintained by beam search, we can substantially speed this up; By grouping together the product between the $W$ matrices and $\boldsymbol{h}_t$ for all $r$ candidates maintained by the beam search procedure, we can use matrix-matrix products that even on the CPU have significantly better cache efficiency. We use the Intel MKL BLAS, and

Table 1: Character-level language model categorical-entropy for the LSTM on AOL search dataset

| Train/test split | Train loss | Validation loss |
|---|---|---|
| Prefix splitting | 1.5454 | 1.4342 |
| Time splitting | 1.5566 | 1.4254 |

the total of these optimizations further reduces the running time to 75ms. By further parallelizing the updates via 8 OpenMP threads brings completion time down to 25 ms.

Finally, one of the most subtle but surprising speedups we attained was through a slightly tweaked LSTM implementation. With the optimizations above, computing the sigmoid terms in the LSTM actually took a surprisingly large 30% of the total computation time. This is due to the fact that 1) our LSTM implementation uses a hard sigmoid activation, which as a clipping operation requires branch prediction; and 2) the fact that the activations we need to apply the sigmoid to are not consecutive in the hidden state vector means we cannot perform fast vectorized operations. By simply grouping together the terms $i_t, f_t, o_t$ in the hidden state, and by using Intel SSE-based operations for the hard sigmoid, we further reduce the completion time down to 13.3ms, or 16.3ms if we include the error correction procedure.

## 5 EXPERIMENTAL RESULTS

We evaluate our method on the AOL search dataset (Pass et al., 2006), a data of real-world searches from 2006. The dataset contains 36M total queries, with 10M of these being unique, illustrating the long tail in these search domains. We set a maximum sequence length for the queries at 60 characters, as this contained 99.5% of all queries.

**Training and testing splits** For each example in the data set, we choose a random cutting point (always after two characters in the string), and treat all text beforehand as the prefix and all text afterwards as the completion. For examples in the test set, we will use these prefixes and actual completions to evaluate the completions that our method predicts. In the training set, we will discard the cutting points and just train on the characters themselves.

We use a test set size of 330K queries, and use the rest for training. We create training and testing splits to evaluate our method using two different strategies:

- Prefix splitting: sort the queries according to the MD5 hash of the prefix then split. This ensures that data in the test set does not contain an exact prefix match in the training set.

- Time splitting: sort the queries by timestamp and split. This mimics making predictions online as new data comes in.

### 5.1 TRAINING LANGUAGE MODEL

We trained our character-level language model on the characters of all the queries in the training set. We trained out LSTM language model for 3 epochs over the entire data set, which took 7.2 hours on a GTX 1080 Ti GPU.

We used a 2-layer LSTM with 256 hidden dimensions with dropout of 0.5 between the two LSTM layers (no dropout within a single layer), and used Adam to train with a step size of 1e-3 and minibatch size of 256. Training and validation losses for the language model, under the two different splittings, are show in Table 1.

We evaluated relatively few other architectures for this model, as the goal here is to use the character-level language model for completion rather than attain state-of-the-art results on language modeling in general. We did, however, find that the LSTM was prone to overfitting if too many hidden units were used, with 512 dimensional hidden units leading to only 1.5235 training loss on the prefix splitting dataset, whereas the validation loss grew to 1.6899.

Table 2: The speedups from different optimizations

| Optimization | Resulting runtime |
|---|---|
| Naive beam search implementation | >1sec |
| Tree-based beam search | 250ms |
| Adding MKL BLAS | 75ms |
| OpenMP parallelization | 25ms |
| Custom LSTM implementation | 13.3ms |
| Adding prefix edit distnace | 16.3 ms |
| Stochastic search | 40 ms |

Table 3: Completion negative log likelihood for stochastic search vs. beam search (lower is better)

| Train/test split | Beam search | Stochastic Search |
|---|---|---|
| Prefix splitting | *2.537* | 3.284 |
| Time splitting | *2.703* | 3.605 |

## 5.2 RUNTIME EVALUATION

Although we mentioned the timing results in the main text, we summarize the speedups achieved by the different optimizations in Table 2, which reports the time to give 16 suggestions for a prefix. One interesting point to note is that stochastic search in this setting actually takes *three times longer* than beam search (to generate the same number of candidates). This is due to the fact that stochastic search tends to generate completions that are much longer than those of beam search, interestingly making the "simpler" method here actually substantially slower while giving worse completions (which we will evaluate shortly).

## 5.3 PERFORMANCE EVALUATION

Finally, we evaluate the actual performance of the completion approaches, both comparing the performance of our beam search method to stochastic search (evaluated by log likelihood under the model), and comparing our completion method to a heavily optimized in-memory trie-base completion model, the standard data structure for completion given string prefixes.

**Stochastic Search vs. Beam Search**   In Table 3 we highlight the performance of beam search versus stochastic search for query completion, evaluated in terms of log likelihood under the model. Although it is not surprising, the results confirm the fact that beam search produces substantially better results under the model likelihood (in addition to being 3x faster, as mentioned above). Note that in this case we are not including any error correction, as it is not trivial to integrate this into the stochastic search setting, and we wanted a direct comparison on sample likelihood.

**Our approach vs. database lookup**   Finally, we compare our total approach (beam search with error correction) to a trie-based (i.e., prefix lookup) completion model. We compare the approach using a combination of two metrics: 1) probabilistic coverage, which is simply the empirical conditional probability of the predicted completion given the prefix

$$\sum_i \hat{P}(\text{completion i} \mid \text{prefix}), \tag{16}$$

where $\hat{P}$ is the empirical probability for the whole AOL dataset (counts); and 2) hit rate, which simply lists the number of times a completion appears in the data set. Because the error correction model adjusts the prefix, it is not possible to compute probabilistic coverage exactly, but we can still get a sense of how likely the completions are based upon how often they occur. Table 4 shows the performance of the trie-based approach, beam search, and beam search with error correction under these metrics. Our models generally outperform trie-based approaches in all settings, the one exception being probabilistic coverage on the time-based training/testing split. This is likely due

Table 4: Performance of our language model based methods versus trie-based prefix lookup.

| Train/test split | Method | Probabilistic coverage | Hit rate |
|---|---|---|---|
| Prefix splitting | Trie-based | 27.5% | 1480 |
| | Beam search | **_39.7%_** | 1575 |
| | Beam search w/ error correction | - | **_3360_** |
| Time splitting | Trie-based | **_48.6%_** | 1273 |
| | Beam search | 31.6% | 1040 |
| | Beam search w/ error correction | - | **_1429_** |

to some amount of shift over time in the search query terms. And although we cannot generate coverage numbers for the error-correction method, the hit rate suggests that it is indeed giving better completions than the alternative approaches.

Further, we note that in addition to these numbers, there are a few notable disadvantages with trie-based lookup. The trie data structure we compare to is very memory intensive (requires keeping prefixes for all relevant queries in memory), and takes a minimum of 16GB of RAM for the entire AOL search data set. The contrasts to the language model approach, which fits in only *18 MB*. And if a prefix has not been seen before in the data set, the trie-based approach will offer no completions. Further, the trie-based approach is not amenable to error correction in isolation, as candidate corrections need to be proposed prior to lookup in the database; the process of repeatedly generating these candidates and performing the lookups will work for at most 2 edits, whereas our approach empirically easily handles completions that include 4-5 edits.

## 6 CONCLUSIONS

In this paper, we presented a search query completion approach based upon character-level deep language models. We proposed a method for integrating the approach with an error correction framework and showed that candidate completions with error correction can be efficiently generated using beam search. We further described several optimizations that enabled the system to work in real time, including a CPU-based custom LSTM implementation. The method is able to jointly produce better completions than simple prefix lookup, while simultaneously being able to generate the candidates in real time.

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
