# OpenReview forum: "Realtime query completion via deep language models"
_ICLR.cc/2018/Conference — Reject_

### Official Review · AnonReviewer2 · 2017-11-19
**Solid engineering but poorly motivated modeling choices for query correction.**

**Rating:** 4
**Confidence:** 5

**Review:**

This paper presents methods for query completion that includes prefix correction, and some engineering details to meet particular latency requirements on a CPU.  Regarding the latter methods: what is described in the paper sounds like competent engineering details that those performing such a task for launch in a real service would figure out how to accomplish, and the specific reported details may or may not represent the 'right' way to go about this versus other choices that might be made.  The final threshold for 'successful' speedups feels somewhat arbitrary -- why 16ms in particular?  In any case, these methods are useful to document, but derive their value mainly from the fact that they allow the use of the completion/correction methods that are the primary contribution of the paper.

While the idea of integrating the spelling error probability into the search for completions is a sound one, the specific details of the model being pursued feel very ad hoc, which diminishes the ultimate impact of these results.  Specifically, estimating the log probability to be proportional to the number of edits in the Levenshtein distance is really not the right thing to do at all.  Under such an approach, the unedited string receives probability one, which doesn't leave much additional probability mass for the other candidates -- not to mention that the number of possible misspellings would require some aggressive normalization.  Even under the assumption that a normalized edit probability is not particularly critical (an issue that was not raised at all in the paper, let alone assessed), the fact is that the assumptions of independent errors and a single substitution cost are grossly invalid in natural language.  For example, the probability p_1 of 'pkoe' versus p_2 of 'zoze' as likely versions of 'poke' (as, say, the prefix of pokemon, as in your example) should be such that p_1 >>> p_2, not equal as they are in your model.  Probabilistic models of string distance have been common since Ristad and Yianlios in the late 90s, and there are proper probabilistic models that would work with your same dynamic programming algorithm, as well as improved models with some modest state splitting.  And even with very simple assumptions some unsupervised training could be used to yield at least a properly normalized model.  It may very well end up that your very simple model does as well as a well estimated model, but that is something to establish in your paper, not assume.  That such shortcomings are not noted in the paper is troublesome, particularly for a conference like ICLR that is focused on learned models, which this is not.  As the primary contribution of the paper is this method for combining correction with completion, this shortcoming in the paper is pretty serious.

Some other comments:

Your presentation of completion cost versus edit cost separation in section 3.3 is not particularly clear, partly since the methods are discussed prior to this point as extension of (possibly corrected) prefixes.  In fact, it seems that your completion model also includes extension of words with end point prior to the end of the prefix -- which doesn't match your prior notation, or, frankly, the way in which the experimental results are described.

The notation that you use is a bit sloppy and not everything is introduced in a clear way.  For example, the s_0:m notation is introduced before indicating that s_i would be the symbol in the i_th position (which you use in section 3.3).  Also, you claim that s_0 is the empty string, but isn't it more correct to model this symbol as the beginning of string symbol?  If not, what is the difference between s_0:m and s_1:m?  If s_0 is start of string, the s_0:m is of length m+1 not length m.

You spend too much time on common, well-known information, such as the LSTM equations.  (you don't need them, but also why number if you never refer to them later?)  Also the dynamic programming for Levenshtein is foundational, not required to present that algorithm in detail, unless there is something specific that you need to point out there (which your section 3.3 modification really doesn't require to make that point).

Is there a specific use scenario for the prefix splitting, other than for the evaluation of unseen prefixes?  This doesn't strike me as the most effective way to try to assess the seen/unseen distinction, since, as I understand the procedure, you will end up with very common prefixes alongside less common prefixes in your validation set, which doesn't really correspond to true 'unseen' scenarios.  I think another way of teasing apart such results would be recommended.

You never explicitly mention what your training loss is in section 5.1.

Overall, while this is an interesting and important problem, and the engineering details are interesting and reasonably well-motivated, the main contribution of the paper is based on a pretty flawed approach to modeling correction probability, which would limit the ultimate applicability of the methods.

---

> ### Author Response · Authors · 2018-01-04
> **Re: comments**
>
> First, we admit that the value of this work is more engineering-oriented, but it successfully solves one of the most important problem to practically use DL for query completion: it reduces the running time from ~ 1 second [Sordoni et al. (2015), table 2] to 16 ms, using only CPU. The reason that the response time threshold is important is that the users always want responsive results, in realtime. Otherwise the query completion won't be that helpful.
>
> Second, let me defense a bit about the error correction model. Because we are doing completion and correction at the same time, the prefix with zero edit won't dominate: the beam search always keeps some different prefixes, and only when the probability became too small will them be kicked out of the candidate set. Essentially, we are only assuming "constant typo penalties" in the prefix; using your example of completing "pokemon":
>
> When the user types "zoze" or "pkoe", the starting log likelihood are both -4*2. But when doing completion, the decrease in log likelihood of "zoze" will be much higher than "pkoe", so it will be kicked out of the candidate set.
>
> Further, say that \log P(pokemon|poke) = -1. When \log P(pkoemon|pkoe)=-30, the beam search process with error correction should choose
> \log P(pokemon|poke) + -4*2 = -9 instead of
> \log P(pkoemon|pkoe) + 0      = -30.
>
> We admit that the model is naive and we should have different penalties in different part of the prefix. But that should be tunable by changing the loss function in the edit distance and we tried the simplest first (and it works). For a demo of the error correction function, you can try it in our online website, but note that we fixed the first two characters in the prefix so the error should happen only after that. We appreciate the comment that we should use a learnt model and welcome references.
>
> Replies to other comments: we modified the notations in the revision. The training loss (categorical entropy) is mentioned in section 2; we made it clear in the revision.
>
> We thank the reviewer again for the helpful comments.

---

### Official Review · AnonReviewer1 · 2017-11-26
**This paper proposes a practical algorithm to solve query completion problem with error correction. The paper is very well written and easy to understand. Experiments show that the algorithm can run in real time on CPU.**

**Rating:** 6
**Confidence:** 3

**Review:**

This paper focuses on solving query completion problem with error correction which is a very practical and important problem. The idea is character based. And in order to achieve three important targets which are auto completion, auto error correction and real time, the authors first adopt the character-level RNN-based modeling which can be easily combined with error correction, and then carefully optimize the inference part to make it real time.

Pros:
(1) the paper is very well organized and easy to read.
(2) the proposed method is nicely designed to solve the specific real problem. For example, the edit distance is modified to be more consistent with the task.
(3) detailed information are provided about the experiments, such as data, model and inference.

Cons:
(1) No direct comparisons with other methods are provided. I am not familiar with the state-of-the-art methods in this field. If the performance (hit rate or coverage) of this paper is near stoa methods, then such experimental results will make this paper much more solid.

---

### Official Review · AnonReviewer3 · 2017-11-29
**Nicely explained, could use more thorough experiments**

**Rating:** 5
**Confidence:** 3

**Review:**

The authors describe a method for performing query completion with error correction using a neural network that can achieve real-time performance. The method described uses a character-level LSTM, and modifies the beam search procedure with a an edit distance-based probability to handle cases where the prefix may contain errors. Details are also given on how the authors are able to achieve realtime completion.

Overall, it’s nice a nice study of the query completion application. The paper is well explained, and it’s also nice that the runtime is shown for each of the algorithm blocks. Could imagine this work giving nice guidelines for others who also want to run query completion using neural networks. The final dataset is also a good size (36M search queries).

My major concerns are perhaps the fit of the paper for ICLR as well as the thoroughness of the final experiments. Much of the paper provides background on LSTMs and edit distance, which granted, are helpful for explaining the ideas. But much of the realtime completion section is also standard practice, e.g. maintaining previous hidden states and grouping together the different gates. So the paper feels directed to an audience with less background in neural net LMs.

Secondly, the experiments could have more thorough/stronger baselines. I don’t really see why we would try stochastic search. And expected to see more analysis of how performance was impacted as the number of errors increased, even if errors were introduced artificially, and expected analysis of how different systems scale with varying amounts of data. The fact that 256 hidden dimension worked best while 512 overfit was also surprising, as character language models on datasets such as Penn Treebank with only 1 million words use hidden states far larger than that for 2 layers. More regularization required?

---

### Public Comment · ~Aaron_Jaech1 · 2017-11-04
**prefix length**

I might have missed this but where do you say what the length of the prefix you used is? I'm assuming you only used a single length prefix based on how you describe doing the train/test split.

---

> ### Author Response · Authors · 2017-11-07
> **Re: prefix length**
>
> For each example in the data set, we choose a random cutting point (always after two characters in the string), as described in the experiment section. That is, we roll a dice for every sample in the testing set to simulate the user inputs (prefixes), which can be of different length. Consider the dataset
>
> google map
> apple
>
> the user might input
>
> google m
> ap
>
> by choosing a random cutting point.

---

### Public Comment · ~Aaron_Jaech1 · 2017-11-18
**metrics**

I have some questions about your metrics.

* In Table 1, why is the validation loss so much better than the training loss? Is that backwards?

* In Table 3, I'm not sure how meaningful these numbers are. The traditional way of evaluating the language model would be to see how much probability it assigns to the true query completion. It seems like what you are doing is generating a completion by sampling from the model and then reporting the probability that the model assigned to it's own sample. The model could be terrible and still assign very high likelihood to whatever sequence it chooses to generate. As you said, obviously, beam search will give a better number than stochastic search.

* In Table 4, you give two metrics: probabilistic coverage and hit rate. If one of the key advantages that you give for your model is that it can generate completions for prefixes that are not found in the training data then it seems you would want a metric that could capture that. My understanding is that probabilistic coverage and hit rate both only apply when the prefix is in the training data. Is that right? Additionally, other papers are query auto-completion on the AOL data seem to use mean reciprocal rank as the main metric. Have you considered using that as a metric as well?

I tried to reproduce your results on my own and was able to confirm that the LSTM LM gives somewhat comparable MRR to previous approaches based on database lookups. I think putting that result in your paper would significantly strengthen your claims.

---

> ### Author Response · Authors · 2018-01-04
> **Re: metrics**
>
> (Table 1) The validation loss is better than training loss because the model is under-fitted.
>
> (Table 3) We are actually sampling the prefix-completion pair from the data instead of from our model. The reason we need to do such sampling is because AOL dataset only consists of whole queries instead of the prefix-completion pair. Thus, we assume that the user may stop typing uniformly and generate the prefix-completion pair by sampling from the data, which is completely independent of our model.
>
> (Table 4)  Both the metrics only apply when the prefix appears in the whole dataset instead of the training data. The prefix for test evaluation might only appears in the test set but not in the training set, but we estimate the empirical probability coverage / hit rate from the whole dataset. For example, if we have the dataset
>
> abc
> abd
> ace
> ...
>
> then the empirical probability for prefix "ab" should be P(abc|ab) = P(abd|ab) = 1/2. While in training, the model might never see the prefix "ab", but the probability coverage metric still work in this case. The reason we separate the probabilistic coverage from hit rate is that if error correction occurs, the prefix (prior) is different and the probability coverage doesn't work, and we must assume typo model to get probabilities. So we show hit rate (counts in the dataset) instead in the case of error correction. MRR also doesn't work in the case because of the data generation process (we don't have the correct user behavior from the AOL dataset). Surely we can "simulate" the typos and create a synthetic dataset, but that would be biased.
>
> Lastly, we also confirm that we have comparable MRR to database lookup when not doing error correction. But we didn't use that because of the reason above.

---

### Decision · Program_Chairs · 2018-01-29
**ICLR 2018 Conference Acceptance Decision**

**Decision:**

Reject

**Comment:**

This paper has some interesting ideas that have been implemented in a rather ad hoc way; the presentation focuses perhaps too much on engineering aspects.